# Patterns and drivers of recent disturbances across the temperate forest biome

Andreas Sommerfeld [1], Cornelius Senf [1,2], Brian Buma [3], Anthony W. D'Amato [4],
Tiphaine Després [5,6], Ignacio Díaz-Hormazábal[7], Shawn Fraver[8], Lee E. Frelich [9], Álvaro G. Gutiérrez [7],
Sarah J. Hart[10], Brian J. Harvey[11], Hong S. He[12], Tomáš Hlásny[5], Andrés Holz [13], Thomas Kitzberger[14],
Dominik Kulakowski [15], David Lindenmayer [16], Akira S. Mori[17], Jörg Müller[18,19], Juan Paritsis [14],
George L. W. Perry [20], Scott L. Stephens[21], Miroslav Svoboda[5], Monica G. Turner [22], Thomas T. Veblen[23] &
Rupert Seidl [1]

Increasing evidence indicates that forest disturbances are changing in response to global change, yet local variability in disturbance remains high. We quantified this considerable variability and analyzed whether recent disturbance episodes around the globe were consistently driven by climate, and if human influence modulates patterns of forest disturbance. We combined remote sensing data on recent (2001–2014) disturbances with in-depth local information for 50 protected landscapes and their surroundings across the temperate biome. Disturbance patterns are highly variable, and shaped by variation in disturbance agents and traits of prevailing tree species. However, high disturbance activity is consistently linked to warmer and drier than average conditions across the globe. Disturbances in protected areas are smaller and more complex in shape compared to their surroundings affected by human land use. This signal disappears in areas with high recent natural disturbance activity, underlining the potential of climate-mediated disturbance to transform forest landscapes.

[1] University of Natural Resources and Life Sciences (BOKU) Vienna, Institute of Silviculture, Peter Jordan Straße 82, 1190 Wien, Austria. [2] Geography Department, Humboldt-Universität zu Berlin, Unter den Linden 6, 10099 Berlin, Germany. [3] Dept. of Integrative Biology, University of Colorado, 1151 Arapahoe, Denver, CO 80204, USA. [4] University of Vermont, Rubenstein School of Environment and Natural Resources, Aiken Center Room 204E, Burlington, VT 05495, USA. [5] Faculty of Forestry and Wood Sciences, Czech University of Life Sciences in Prague, Kamýcká 129, 165 21 Prague 6, Czech Republic. [6] Institut de Recherche sur les Forêts, Université du Québec en Abitibi-Témiscamingue, 445 boulevard de l'Université, Rouyn-Noranda, QC J9X 5E4, Canada. [7] Facultad de Ciencias Agronómicas, Departamento de Ciencias Ambientales y Recursos Naturales Renovables, Universidad de Chile, Av. Santa Rosa 11315, La Pintana, 8820808 Santiago, Chile. [8] University of Maine, School of Forest Resources, 5755 Nutting Hall, Orono, Maine 04469, USA. [9] Department of Forest Resources, University of Minnesota, 1530 Cleveland Ave. N., St.Paul, MN 55108, USA. [10] Department of Forest and Wildlife Ecology, University of Wisconsin–Madison, Madison, WI 53706, USA. [11] School of Environmental and Forest Sciences, University of Washington, Seattle, WA 98195, USA. [12] School of Geographical Sciences, Northeast Normal University, Changchun 130024, China. [13] Department of Geography, Portland State University, Portland, OR 97201, USA. [14] INIBIOMA, CONICET-Universidad Nacional del Comahue, Quintral 1250, Bariloche, 8400 Rio Negro, Argentina. [15] Clark University, Graduate School of Geography, Worcester, MA 01602, USA. [16] Fenner School of Environment and Society, The Australian National University, Canberra, ACT 2601, Australia. [17] Graduate School of Environment and Information Sciences, Yokohama National University, Yokohama 240-8501, Japan. [18] Field Station Fabrikschleichach, Department of Animal Ecology and Tropical Biology, Biocenter, University of Würzburg, Glashüttenstraße 5, 96181 Rauhenebrach, Germany. [19] Bavarian Forest National Park, Freyunger Str. 2, 94481 Grafenau, Germany. [20] School of Environment, University of Auckland, Auckland 1142, New Zealand. [21] Department of Environmental Science, Policy and Management, University of California, Berkeley, CA 94720, USA. [22] Department of Integrative Biology, Birge Hall, University of Wisconsin–Madison, Madison, WI 53706, USA. [23] Department of Geography, University of Colorado, Boulder, CO 80309, USA. These authors contributed equally: Andreas Sommerfeld, Cornelius Senf. Correspondence and requests for materials should be addressed to A.S. (email: andreas.sommerfeld@boku.ac.at)

Natural disturbances are an essential component of forest ecosystems[1]. Yet, forest disturbance regimes are changing in response to global climate change[2]. Hotter and prolonged droughts[3], exceptional bark beetle outbreaks[4], and megafires[5] have been increasingly reported in recent years, and are impacting forest ecosystems across all forested continents[2]. Changes in disturbance dynamics can have substantial impacts on ecosystem services provided by forests, e.g., climate regulation[6,7], provisioning of drinking water[8,9], and protection from natural hazards[10], as well as affect conservation of biological diversity[11]. Quantifying disturbance patterns (i.e., the size, shape, and prevalence of disturbances in forest landscapes) and understanding their drivers is thus a key challenge for ecological research.

While the potential drivers of the ongoing disturbance change are global, the responses to these drivers vary considerably at the local scale. Insights from well-studied systems such as Yellowstone National Park in North America[1], the Bohemian Forest ecosystem in Europe[12], and the O'Shannassy water catchment in Australia[13] have provided important insights into the complex interactions between climate variability, disturbances, and forest development. While an in-depth understanding of disturbance dynamics exists for a growing number of landscapes (i.e., contiguous land areas of roughly between $10^3$ and $10^6$ ha) around the globe, their responses to global drivers have not consistently been compared to date. Questions such as whether recent bark beetle outbreaks in North America differ from those in Europe with regard to their climate sensitivity, or whether recent fires in Australia created similar patterns as those in the Americas remain largely unexplored. Comparing the variation in disturbance patterns and their relationship to climate variability among landscapes at subcontinental to global scales[14,15] has the potential to elucidate whether recent disturbance episodes were consistently driven by climate across continents, or whether climate sensitivities differ between systems. Furthermore, such a comparison can shed light on how regional-to continental-scale drivers such as climate variability interact with

local factors such as the topographic template of a landscape[16] and the influence of human land use[17]. A better understanding of global disturbance patterns and their multi-scale drivers is also crucial for improving the representation of disturbances in global vegetation models[18,19], and can have important implications for policy decision making, e.g., in the context of climate change mitigation[20,21].

New opportunities for global forest disturbance research arise from recent advances in remote sensing. Increasingly available remotely sensed datasets on forest disturbance[22] offer high spatial resolution and are globally consistent, enabling large-scale comparative efforts. However, while the global mapping of forest disturbances is now feasible[23,24], attributing disturbance agents from remote sensing data and distinguishing between natural and anthropogenic disturbances remains challenging[25]. Furthermore, ecological context information such as the prevailing tree species composition cannot usually be gleaned from space, underlining the importance of terrestrial information and local ecosystem understanding for a meaningful interpretation of remotely sensed disturbance information. Integrating remote sensing analyses with in-depth knowledge on selected ecosystems across the globe can provide new insights by combining the consistent synoptic view of satellite analysis with the expertise and insights gained from decades of local forest disturbance research.

Our objective was to analyze patterns and drivers of recent disturbances across temperate forests at the global scale. We compiled a global network of 50 protected forest landscapes each with > 2000 ha contiguous forest area (Fig. 1) for which in-depth local systems knowledge exists. We jointly analyzed severe canopy disturbances (i.e., complete mortality of all trees taller 5 m within a 30 m grid cell) in these landscapes using Landsat-derived disturbance maps for 2001–2014[24]. Focusing our network of landscapes on protected areas and their surroundings allowed us to isolate patterns of natural disturbances (inside protected areas) from those of areas where natural and human disturbances

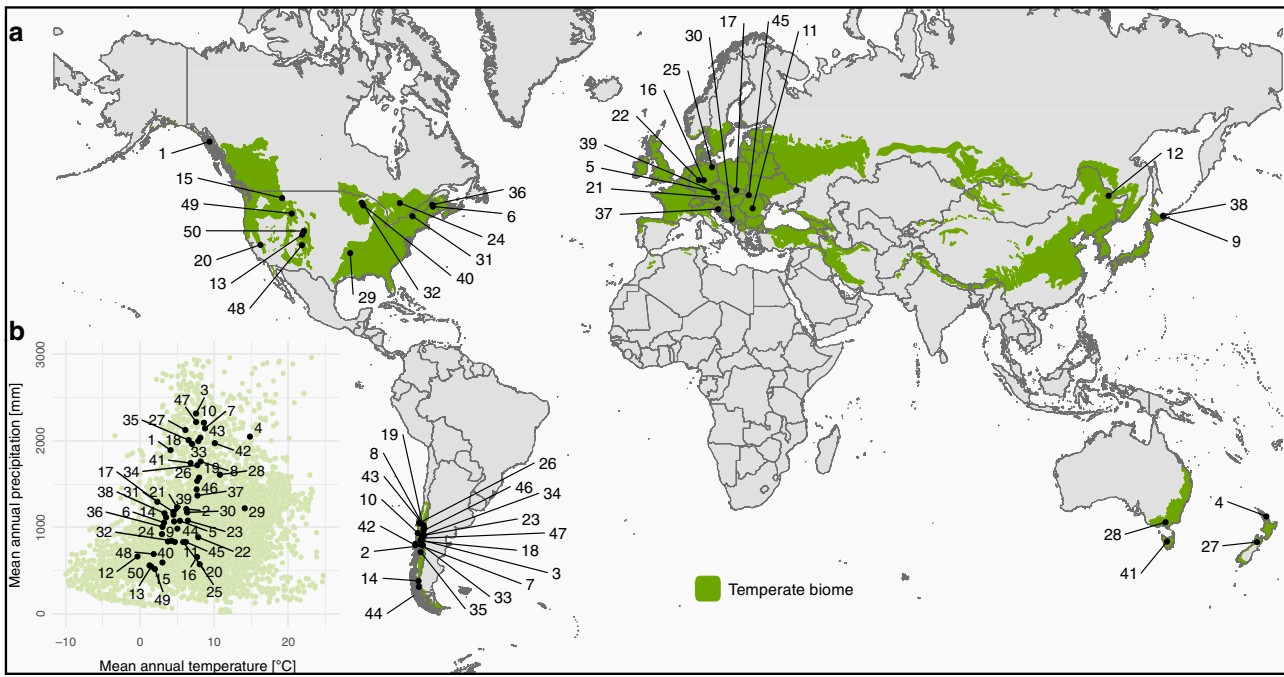

**Fig. 1** A network of 50 protected landscapes to understand global patterns and drivers of temperate forest disturbances. **a** The geographic location of the landscapes, and **b** their location in climate space. The area of the temperate biome is indicated in green[47]. See Supplementary Table 1 for more detailed information on the individual landscapes. Note that the climatic envelope of the biome (green dots in **b**) is based on a sample of 10,000 4500 m×4500 m grid cells throughout the biome

**Table 1 Characteristics of disturbance clusters**

| Cluster | Low | Moderate | High |
|---|---|---|---|
| Number of landscapes | 18 | 23 | 9 |
| Total forest area [ha] | 788,986 | 1,216,364 | 1,965,572 |
| Mean annual temperature [°C] | 6.5 (5.6-7.5) | 5.3 (4.4-6.2) | 3.7 (2.8-4.6) |
| Mean annual precipitation [mm] | 1393 (1241-1544) | 1222 (1071-1374) | 1197 (1046-1349) |
| Mean percent of forest area disturbed 2001–2014 [%] | 0.31 (0.13-0.48) | 4.61 (0.44-8.79) | 21.50 (13.86-29.18) |
| Edge density [m/ha] | 2.87 (1.24-4.50) | 21.69 (2.80-40.58) | 43.22 (25.53-60.91) |
| Area-weighted mean patch size [ha] | 0.66 (0.46-0.85) | 24.22 (6.96-41.47) | 4451.04 (365.24-8536.84) |
| Area-weighted mean perimeter-area-ratio [m/ha] | 960.09 (905.26-1014.92) | 617.28 (560.74-673.82) | 215.31 (150.15-280.74) |

Characteristics of three global clusters of disturbance activity, determined based on satellite-derived disturbance metrics using Gaussian finite mixture models. Values in parentheses indicate the 95% confidence interval

interact (outside protected areas). We concentrated our analysis on a single biome as we were particularly interested in within-biome variation in disturbance patterns and drivers, rather than the comparatively well-studied between-biome differences. We selected temperate forests as the target of our study as they are affected by a wide variety of disturbance agents, frequently contain both angiosperm and gymnosperm species (i.e., high trait variability), and are represented in both the northern and southern hemisphere, spanning an extensive gradient in environmental conditions.

We hypothesized that differences in disturbance agents and tree species are the main determinants of among-landscape variation in disturbance patterns across the globe (H1). Specifically, we expected that areas predominately affected by fire as well as landscapes dominated by tree species with high general susceptibility (i.e., traits such as high maximum tree height and low wood density) are most affected by disturbances. Our alternative hypothesis was that spatial proximity of landscapes is a good indicator of similarities in disturbance patterns, with global differences in disturbances mainly explained by geographical location (e.g., on different continents or hemispheres). To test these hypotheses, we calculated four disturbance indicators, whereof two are landscape-level metrics (percent of landscape disturbed 2001–2014, edge density) and two are patch-level metrics (area-weighted mean patch size, area-weighted mean perimeter-area-ratio), and used cluster analysis to identify patterns among landscapes indicating differences in recent disturbance activity (with disturbance activity jointly referring to the prevalence, size, and shape of disturbances as characterized by our four focal indicators). Furthermore, we hypothesized that the patterns of natural disturbances (i.e., those observed in protected areas with only minimal direct human influence) differ from the combined natural and human disturbances outside protected areas (H2). We expected disturbances in protected areas to be generally smaller and more complex in shape (i.e., higher perimeter-area-ratio) compared to those outside of protected areas. This hypothesis is based on the insights that disturbances outside protected areas are the result of both natural and human disturbances (which can amplify each other, e.g., when strong winds uproot edge trees of freshly created clear-cuts), and that management-related biotic homogenization has the potential to increase forest susceptibility to disturbances relative to natural ecosystem development[26–28]. The alternative hypothesis was that high recent levels of natural disturbance activity supersede the signal of human land use, with similar disturbance patterns inside and outside protected areas. Finally, based on local and regional studies highlighting the importance of climate variability[29] and landscape structure[30] for disturbance dynamics, we tested for a consistent global relationship among climate variability, relative topographic complexity, and the spatio-temporal dynamics of forest disturbances. If recent disturbance episodes are responding consistently to climatic and topographic drivers, we would expect to find a non-random signal in a regression analysis across our set of globally distributed landscapes (H3). Alternatively, if responses vary among landscapes and cancel each other out at the global scale, the regression coefficients for these drivers would not differ significantly from zero.

## Results

**Patterns of recent natural disturbances.** Disturbance dynamics between 2001 and 2014 varied strongly across the temperate forest biome. Unsupervised cluster analysis identified three distinct groups of landscapes based on their recent disturbance dynamics (Table 1; Supplementary Figure 1), which we in the following refer to as low, moderate, and high disturbance activity clusters. Each cluster comprises a group of landscapes of similar characteristics with regard to the size and shape of disturbance patches, the percentage of a landscape disturbed during the study period, and the average amount of edges created by these disturbances (Supplementary Figure 2). Approximately one-third of the landscapes studied (representing 19.9% of the forest area) fell within the low disturbance activity cluster. This group was characterized by small and complex disturbance patches (Table 1), with disturbances on average affecting only 0.31% of the landscape's forest area between 2001 and 2014. Examples of landscapes with low disturbance activity are the Te Paparahi Conservation Area (New Zealand), Shiretoko (Japan), Feng Lin (China), Five Ponds (USA), Hainich (Germany), and Hornopirén (Chile). The majority of the landscapes (23 landscapes, representing 30.6% of the forest area studied) fell within the moderate disturbance activity cluster. This group was characterized by a roughly 30 times larger area-weighted mean patch size than the landscapes in the low disturbance activity cluster (Table 1). Disturbance patches in the moderate cluster were less complex (i.e., had a lower area-weighted mean perimeter-area-ratio) but affected a larger forest area (on average 4.61% of the landscape) between 2001 and 2014. Examples of landscapes in this group are the Bavarian Forest (Germany), Baxter State Park (USA), Los Alerces (Argentina), and Nelson Lakes National Park (New Zealand). Although only 9 of the 50 landscapes analyzed fell within the high disturbance activity cluster, they accounted for 49.5% of the total forest area under study. Area-weighted mean patch size was two orders of magnitude larger than in the moderate disturbance activity cluster, and disturbance patches were considerably less complex (Table 1). On average, almost one quarter of the landscape's forested area was affected by disturbances between 2001 and 2014 in the high disturbance activity cluster, resulting in the highest edge density among all three

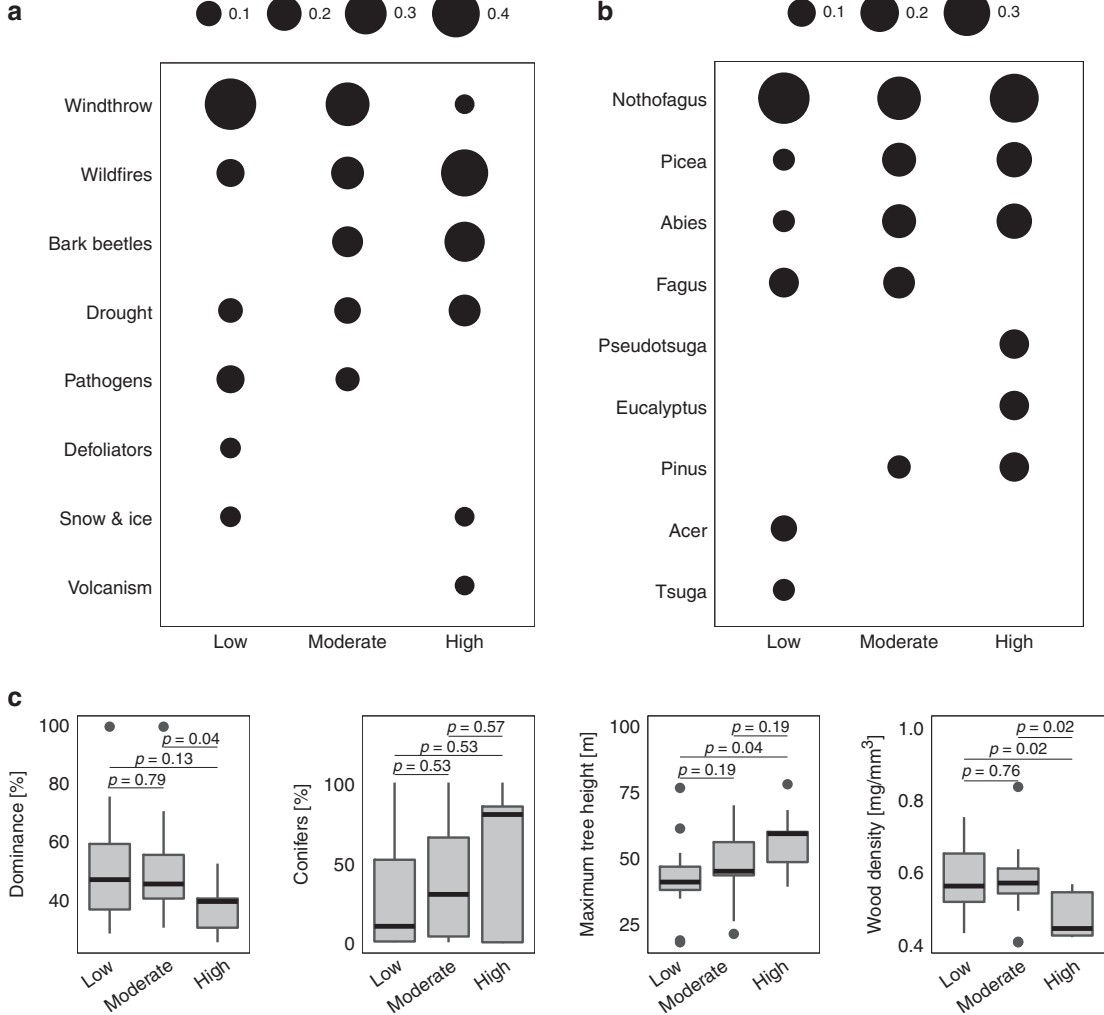

**Fig. 2** Distribution of disturbance agents, tree genera, and tree species traits across three global clusters of disturbance activity (cf. Table 1). Bubbles are scaled relative to the occurrence of the two most important **a** disturbance agents and **b** tree genera within each cluster. **c** Dominance [%] indicates the share of the single most prevalent tree species on the overall tree species composition, while conifers [%] indicates the respective share of all conifer species. Maximum tree height and wood density indicate a weighted trait distribution across landscapes in the respective disturbance activity clusters. Boxplots denote the median (center line) and interquartile range (box), with whiskers extending to three times the interquartile range and points indicating values outside this range. Test statistics and p-values are based on approximate Kruskal–Wallis tests with 9999 permutations. For further information on statistical analyses see Supplementary Table 2

clusters. Examples for landscapes in this group are Yellowstone (USA), Puyehue (Chile), and O'Shannassy (Australia).

The clustering based on the four landscape metrics considered here did not reveal strong geographical patterns (Supplementary Figure 3a). Landscapes from at least three continents were present in each cluster, and all three disturbance activity clusters were represented in both the southern and northern hemispheres. Furthermore, no clear pattern emerged when comparing the clusters in climate space (Supplementary Figure 3b), although a tendency of cooler landscapes experiencing higher disturbance activity could be detected (Table 1).

Disturbance activity clusters were associated with different major disturbance agents and tree genera (Fig. 2a, b, Supplementary Figure 4). Agents differed significantly among disturbance activity clusters ($\chi^2 = 37.64$, $p < 0.01$). Landscapes in the low disturbance activity cluster were frequently affected by multiple disturbance agents, with windthrow being the most prevalent agent. Major bark beetle outbreaks were largely absent

from this group of landscapes. In landscapes experiencing moderate disturbance activity in 2001–2014, fire and bark beetle outbreaks were more prevalent compared to the low cluster. High disturbance activity was predominately associated with wildfire, with bark beetle outbreaks and drought also being important agents in highly disturbed landscapes.

Dominant tree genera (i.e., the genera with the highest proportion of basal area) differed among disturbance activity clusters ($\chi^2 = 69.09$, $p < 0.001$). Low disturbance activity landscapes were largely dominated by broadleaved trees from the genera *Nothofagus*, *Fagus*, and *Acer*, i.e., species that have a relatively low maximum attainable height but higher wood density (Fig. 2c). The moderate disturbance activity cluster was characterized by both broadleaved and coniferous tree species, yet their average trait characteristics largely resembled those of the low disturbance activity landscapes. High disturbance activity landscapes in the northern hemisphere were dominated by the genera *Picea, Abies, Pseudotsuga,* and *Pinus*, which are

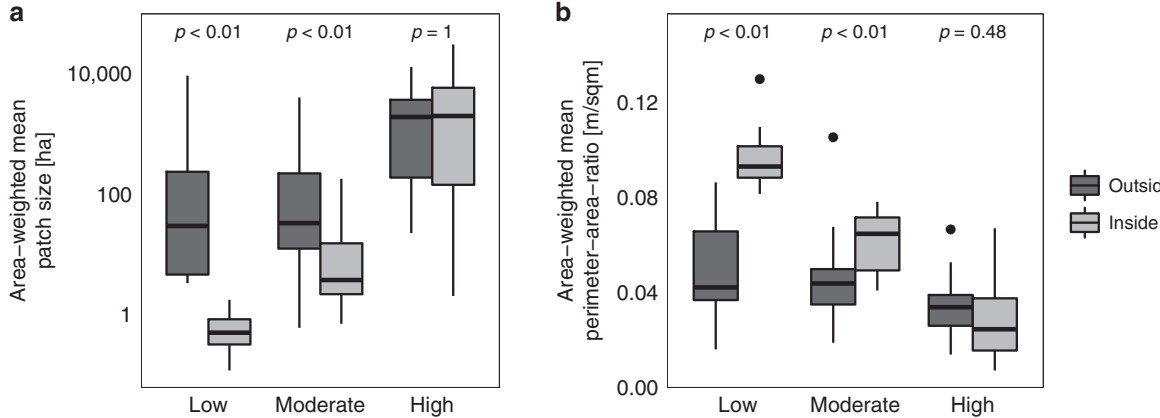

**Fig. 3** Comparison of disturbance patterns inside and outside protected areas. **a** Area-weighted mean patch size and **b** area-weighted mean perimeter-area-ratio are compared for areas inside and outside protected areas for three global clusters of disturbance activity (cf. Table 1). Boxplots denote the median (center line) and interquartile range (box), with whiskers extending to three times the interquartile range and points indicating values outside this range. Test statistics and *p*-values are based on approximate Kruskal–Wallis tests with 9999 permutations

characterized by a higher maximum attainable tree height and lower wood density (see Supplementary Table 2 for test statistics on trait differences between clusters). Conversely, the high disturbance activity landscapes located in the southern hemisphere were mainly characterized by *Nothofagus*. The share of the single most dominant tree species on the overall species composition did not differ between landscapes in the low and moderate clusters. In high disturbance activity landscapes, however, the most important tree species was less dominant compared to landscapes in low and moderate clusters (Fig. 2c).

**Disturbance differences inside and outside protected areas.** Disturbance patches inside protected areas—almost exclusively influenced by natural disturbance agents (but see Supplementary Table 6)—were smaller and more complex than disturbance patches in surrounding areas affected by both human and natural disturbances in the low and moderate disturbance activity clusters (Fig. 3, Supplementary Table 3). For landscapes with high disturbance activity, however, the distribution of patch sizes and perimeter-area-ratios did not differ significantly between protected areas and their surroundings. Hence, there is a higher similarity between disturbances in protected and unprotected systems in areas that experienced high disturbance activity recently. Furthermore, patch size and patch complexity differed more strongly among disturbance activity clusters in protected systems compared to unprotected systems.

**Drivers of spatio-temporal disturbance dynamics.** Inter-annual climate variability was an important driver of temporal disturbance dynamics in all three disturbance activity clusters. The full model (including both temperature/precipitation anomalies and topographic complexity as predictors) was more strongly supported by the data than the spatial-only model (including only topographic complexity) and the Null model (likelihood-ratio tests; all *p*-values < 0.01; see Supplementary Table 4). However, the direction and strength of effects, as well as the lag time of climate effects, varied among clusters (Supplementary Figure 5; Fig. 4). A 2-year and 3-year lag was most strongly supported by the data for the low and moderate disturbance activity clusters, respectively. Conversely, climate variability had an immediate influence on disturbances (i.e., zero lag) in the high disturbance activity cluster (Supplementary Figure 5). For landscapes with low disturbance activity, we found a significant but moderately

negative effect of temperature anomaly (GLMM; $\beta = -0.20$, std. error $= 0.07$, $p = 0.01$; see Supplementary Table 4). That is, warmer than average temperatures decreased the probability of disturbance in the following years (Fig. 4a). This effect of temperature was independent of variation in precipitation. Precipitation anomalies had a significant positive direct effect on disturbance probability (GLMM; $\beta = -0.33$, std. error $= 0.07$, $p < 0.01$; Supplementary Table 4), with wetter conditions increasing disturbance probability in the low disturbance activity cluster (Fig. 4a). Conversely, for landscapes in the high disturbance activity cluster, we found a significant positive effect of temperature (GLMM; $\beta = 0.59$, std. error $= 0.01$, $p < 0.01$; Supplementary Table 4), with a higher disturbance probability in years with above average temperature (Fig. 4c). This effect was further modulated by precipitation, and was strongly amplified if warm years coincided with drier than average conditions (GLMM; $\beta = -0.43$, std. error $< 0.01$, $p < 0.01$; Supplementary Table 4). While landscapes in the moderate disturbance activity cluster showed an overall negative effect of temperature on disturbance probability (GLMM; $\beta = -0.30$, std. error $= 0.02$, $p < 0.01$; Supplementary Table 4), the effect was also significantly influenced by precipitation (GLMM; $\beta = -0.17$, std. error $= 0.03$, $p < 0.01$; Supplementary Table 4). In addition, disturbance probability increased following years with above average temperature and below average precipitation (Fig. 4b).

Relative topographic complexity significantly affected spatial disturbance patterns, with different effects across disturbance activity clusters (Supplementary Table 4). In the predominately wind-influenced low disturbance activity landscapes, disturbance probability increased by ~7.5% with an increase in topographic complexity by one standard deviation (GLMM; $\beta = 0.30$, std. error $= 0.04$, $p < 0.01$; Supplementary Table 4). In contrast, disturbance probability decreased by ~1.25% (GLMM; $\beta = -0.05$, std. error $= 0.01$, $p < 0.01$; Supplementary Table 4) and 2.5% (GLMM; $\beta = -0.10$, std. error $< 0.01$, $p < 0.01$; Supplementary Table 4) with the same amount of change in topographic complexity in the mainly beetle-driven and fire-driven landscapes of the moderate and high disturbance activity clusters.

**Discussion**

We present a quantitative analysis of recent disturbance dynamics across the temperate forest biome. Variation in recent forest disturbance activity across the globe was considerable, spanning a

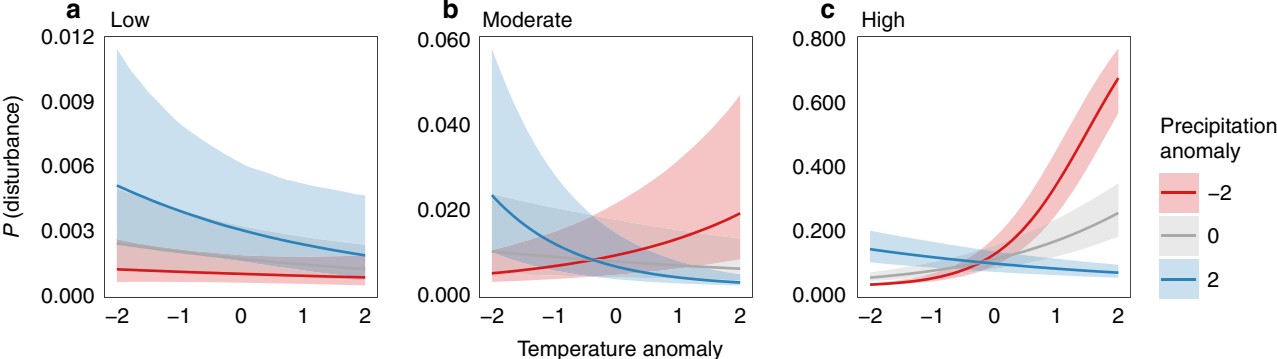

**Fig. 4** Predicted response of disturbance probability to temperature anomaly, modulated by precipitation anomaly. **a–c** The climate sensitivity of disturbances separately for three global clusters of disturbance activity (cf. Table 1). Anomaly values are units of standard deviation with zero indicating the long-term mean. Y-axes are scaled differently across the three panels for clarity of presentation. Prediction uncertainty was estimated from 9999 model simulations, with the lower and upper limit representing the 2.5 and 97.5% quantile of all simulations. We note that prediction intervals include both parameter and model uncertainty, and overlapping prediction intervals can occur despite significant differences in parameter values. For parameter estimates and standard errors, see Supplementary Table 4. The number of experimental replicates equals the number of study sites per cluster (Low: 18, Moderate: 23, High: 9)

large gradient of patch sizes and landscape area affected by disturbance. Our results highlight that while extensive disturbances such as massive bark beetle outbreaks or severe large-scale fires have garnered considerable attention from researchers and the public recently, many temperate forests are dominated by small-scale disturbance events. This finding underscores the importance of a consistent quantification and analysis of disturbance dynamics across systems. The main disturbance agent affecting a system was more indicative of within-biome variation in disturbance activity than geographical proximity of landscapes or their location on the same continent or hemisphere (H1). Specifically, wind was an important agent responsible for small-scale disturbances in temperate forests[31]. Wildfires and bark beetle outbreaks, on the other hand, were the two most prominent agents associated with large and severe disturbances in recent years. However, the fact that both wind and wildfires occurred in all three clusters of disturbance activity highlights the considerable within-biome variability even within disturbance regimes characterized by the same agent.

Tree species composition was related to global differences in recent disturbances, with Northern Hemisphere temperate forests dominated by conifers experiencing elevated disturbance activity. This pattern can partly be explained by the general life-history strategy of conifers and their extensive coevolution with disturbances[32], as many conifers are well adapted to either tolerate disturbances or swiftly recolonize disturbed areas. An unexpected finding was that the dominance of the single most prevalent tree species was lower in landscapes with high disturbance activity compared to those with low and moderate disturbance activity. This contrasts with previous suggestions that disturbance risk increases if landscapes are dominated by a small number of tree species[33]. However, disturbances themselves can have a positive effect on tree species diversity[34,35]. Consequently, higher evenness in systems strongly affected by disturbances—as found here— could be a consequence of disturbances, rather than being causally related to them.

In large parts of the temperate forest biome, human disturbances dominate the landscape. Consequently, the disturbance patterns of unprotected systems differed from the natural disturbance regime observed in protected areas. In the majority of human-dominated landscapes outside protected areas, disturbance patches were larger and less complex than in protected

areas, supporting our expectation (H2). In landscapes affected by large-scale disturbances, however, the patterns in protected areas and their surroundings were similar. This suggests that large natural disturbances can override the effect of human land use and dominate landscape patterns in forest ecosystems. As these events have been found to be particularly climate sensitive (Fig. 4c), future conditions could produce more coarse-grained landscape patterns (i.e., landscapes characterized by larger patch sizes) in temperate forests[36].

An important caveat associated with our analysis is that we could not consider vegetation structure and disturbance history as potential covariates in our analyses, resulting from the lack of a globally consistent data set on these variables. It therefore remains unclear whether the differences between protected and unprotected areas arise from a higher disturbance susceptibility of the latter systems, or whether they reflect management activities such as clearcutting in areas outside protected landscapes. Disturbance history and ecological legacies can exert an important influence on current disturbance activity[37,38]. This is important as most of our study landscapes have been protected only for a few decades, and legacies from former land use might still persist. Furthermore, although our 50 protected landscapes cover the climatic envelope of temperate forests well (Fig. 1b), they are not necessarily representative of the full range of ecological conditions of the entire temperate forest biome. Selecting areas with a long history of forest dynamics research enabled us to consistently compare patterns and responses across locally well-researched systems, but might also be a source of bias that should be considered in interpreting our results. The finding of similar disturbance patterns inside and outside of protected areas in the high disturbance activity cluster may, for instance, partly result from the generally remote location of these particular landscapes, with limited human activity in their surroundings. Future analyses explicitly contrasting disturbances inside and outside protected areas could help to better understand the effect of human activity on disturbances, and quantify the impact of anthropogenically altered disturbance regimes on biodiversity[25,39]. Furthermore, we here focused on severe canopy disturbances with complete canopy mortality of all trees taller 5 m within a 30 × 30 m grid cell[24]. Consequently, low severity disturbances and understory tree mortality were not considered, potentially leading to an underestimation of forest disturbance activity. However, the

severe canopy disturbances examined here are generally well represented by the data set used: For example, Buma and Barrett[40] found an overall agreement of 91% when comparing the global disturbance data set used here to high-resolution imagery in southern Alaska. Furthermore, Borelli et al.[41] determined an overall accuracy of 81% in an independent evaluation of the data across Europe. We are thus confident that the data set used here is able to capture the variability in severe canopy disturbances across the temperate forest biome.

A remaining limitation of our analysis is the relatively short duration of our study period. The currently available disturbance time series from satellite data remain too short to characterize disturbance regimes[1] satisfactorily, and preclude the assessment of temporal trends in disturbance[7,42]. Consequently, we focused only on the effect of inter-annual climate variability rather than on long-term trends, using temperature and precipitation anomalies as predictors. Future work should also consider the effect of climatic extremes and intra-annual climate patterns for refining our understanding of climate—disturbance relationships[29]. In addition, process-based simulation modeling[43] could be employed to obtain a more dynamic and long-term perspective on global disturbance regimes and their responses to a changing climate.

Here we provide evidence that high recent disturbance activity in temperate forest ecosystems across the globe was strongly related to the joint occurrence of warmer and drier than average climate conditions (H3). These global scale findings are in general agreement with local studies[4,14,42,44,45], particularly considering that our high disturbance activity cluster was dominated by wildfires, bark beetle outbreaks, and drought. Our results therefore suggest that a warming climate could facilitate large-scale disturbances in temperate forest ecosystems in the future[2,46]. Our findings also show that climate sensitivity can, to some degree, be buffered by heterogeneous topography, which impedes the spread of disturbances and/ or increases the complexity of disturbed patches[25]. Interestingly, our analysis suggests that both the effect of climate and the effect of topography reversed for landscapes characterized by low disturbance activity, compared to those with high disturbance activity. For the former, which are predominately driven by wind disturbance, cooler and wetter conditions as well as higher relative topographic complexity increased disturbance probability[31], which is consistent with decreased tree anchorage (soil wetness) and increased wind susceptibility (exposed ridges, funnel effects) under such conditions. However, the signals detected for low disturbance activity areas were generally weak, underlining the highly stochastic nature of small-scale mortality events in forest ecosystems.

We conclude by emphasizing the importance of protected areas for understanding changes in forest landscapes in the absence of direct human influences. Furthermore, our work highlights the importance of consistent global information for characterizing patterns and identifying drivers of important ecological processes such as disturbances. Quantitative baselines acknowledging the substantial spatio-temporal variability in ecosystems are needed to identify, monitor, and attribute changes in ecological processes. The analyses presented here are an important step towards such an improved quantitative characterization of forest disturbances at the global scale, combining large-scale remote sensing data with ecological context information from local experts. An improved quantitative characterization of forest disturbances at the global scale can, for instance, inform the development and application of global vegetation models, which largely ignore the impacts of disturbances to date, or only consider a highly simplified representation of disturbance processes[18,19]. An improved consideration of disturbance processes in future projections is important as our results highlight the considerable sensitivity of disturbances to the ongoing changes in the climate system. We conclude that the

resilience and adaptive capacity of ecosystems to disturbances remain important priorities of forest research and management.

## Methods

**A biome-wide network of protected forest landscapes.** We compiled a network of study landscapes distributed throughout the temperate forest biome as defined by Olson et al.[47]. (see also Fig. 1). Selection criteria were that the landscapes are protected (i.e., IUCN Cat. I and Cat. II), and have a minimum of 2000 ha of contiguous forest area. Studying protected areas allowed us to largely control for anthropogenic disturbances and focus our main analyses on natural disturbances. We analyzed 50 landscapes distributed across 16 countries on five continents, representing a forest area of 3.9 Mill. ha (median landscape size: 30,889 ha; see Supplementary Table 1 for details). The study landscapes cover a wide climatic gradient of the temperate forest biome, with mean annual temperatures ranging from −0.3 °C to 14.8 °C, and mean annual precipitation sums between 517 mm and 2315 mm (Supplementary Table 1 and Fig. 1b).

**Disturbance data and landscape pattern analysis.** We acquired forest cover and annual disturbance maps (2001–2014) from Hansen et al.[24] (Version 1.2) at 30 m spatial resolution. A disturbance was defined as a severe canopy disturbance, meaning the complete mortality of all trees taller 5 m within a pixel[24]. Only disturbance events that occurred between 2001 and 2014 were considered. To characterize disturbance patterns, we calculated two landscape-level metrics and two patch-level metrics for each study landscape, using an eight-neighbor rule for defining adjacency and considering disturbances throughout the entire study period. The landscape-level metrics were: (i) percent of landscape disturbed 2001–2014, and (ii) edge density of all disturbed patches within the forest area of a landscape; with the patch-level metrics being (iii) area-weighted mean patch size, and (iv) area-weighted mean perimeter-area-ratio. To identify differences and similarities in recent disturbance patterns across the temperate forest biome, we used Gaussian finite mixture models, as implemented in the R package *mclust*[48] (version 5.4). Gaussian finite mixture models are an approach for unsupervised clustering, used here to identify groups of landscapes with similar disturbance patterns. The optimal number of cluster centers was determined by maximizing the Bayesian Information Criteria (BIC). Robust statistics for characterizing each cluster were derived by using parametric bootstrapping (9999 replications). Subsequently, the clusters were characterized with regard to their main disturbance agents and tree genera (i.e., the two most important agents and genera for a landscape during the period 2001–2014; see below for details). To describe potential functional differences between the clusters, we included plant traits in our analysis. After an initial screening and analysis of multicollinearity, we focused on two complementary plant traits corresponding to disturbance resistance and susceptibility, i.e., maximum potential tree height and mean wood density, extracted from the TRY database[49]. Plant height directly increases susceptibility to wind disturbance and is also a proxy for biomass accumulation potential, which is related to fuel load in the context of disturbances by wildfire. Wood density is positively correlated with the ability of trees to resist physical forces such as wind and drought. For each landscape, weighted trait means based on tree species shares were calculated. Differences in agents and tree genera among clusters were tested using approximate Pearson $\chi^2$ tests of homogeneity with 9999 permutations. Differences in traits among clusters were tested using two-tailed pairwise approximate Kruskal–Wallis tests with 9999 permutations, applying false discovery rate correction. All test procedures were used as implemented in the *coin*[50] package (version 1.2-1) in R[51].

**Expert-based information on local ecological context.** Remote sensing provides a consistent estimate of disturbances across the biome, yet the ecological context of these disturbances, such as the dominant disturbance agents and the tree species affected, cannot be inferred from space. We thus consulted local experts for all 50 landscapes, collecting ecological context information via a questionnaire (see Supplementary Table 5). The questionnaire included four questions, two of which were multiple choice questions with the opportunity to add additional answers. They focused on determining the dominant tree species, the main disturbance agents affecting a landscape between 2001 and 2014, and the impact of disturbances on particular tree species. Values of tree species dominance in a given landscape were estimated as basal area shares. Local experts were identified via their publication record on the topic of forest dynamics for the selected areas. All consulted experts also contributed to the analysis and interpretation of the data, and are identified individually in Supplementary Table 1.

**Difference inside and outside protected areas.** After focusing on natural disturbance dynamics in protected areas in the previous analysis steps, we asked how protected areas differed from the disturbance dynamics in the unprotected systems surrounding our study landscapes. To compare spatial disturbance patterns inside vs. outside protected areas, we extracted all forest disturbances in a buffer surrounding the protected landscapes. The buffer size was selected proportional to the landscape size, and was set to the diagonal of the minimum bounding rectangle of each study landscape. We compared area-weighted mean patch size and area-weighted mean perimeter-area-ratio between strata (i.e., inside vs. outside

protected areas) using boxplots, and tested differences using two-tailed approximate Kruskal–Wallis tests with 9999 permutations.

**Drivers of spatio-temporal disturbance dynamics.** We used generalized linear mixed effects models[52] (GLMMs) to test the influence of relative topographic complexity and climatic variability on spatial and temporal disturbance dynamics. Analyses were run separately for each disturbance activity cluster, modeling the annual probability of disturbance at the pixel level. As response variable we used annual binary disturbance maps indicating whether a pixel was disturbed in a given year or not. Consequently, the GLMMs were specified with a binomial error distribution and a logit link-function. As measure of topographic complexity we used the topographic ruggedness index[53] (TRI), which was calculated from a 30 m digital elevation model obtained from the Shuttle Radar Topographic Mission (SRTM). We calculated the TRI using a window size of $7 \times 7$ pixels, depicting the topographic complexity within a radius of ~100 m around a focal pixel. TRI values were subsequently scaled to zero mean and a standard deviation of one for each landscape, with negative values indicating a lower than average relative topographic complexity, and positive values indicating a higher than average relative topographic complexity. As a measure of climate variability we obtained time series of mean annual temperature and annual precipitation sum from *FetchClimate*[54], which are based on daily climate data from the NCEP/NCAR Reanalysis 1 database. We calculated climate anomalies by scaling the time series to zero mean and a standard deviation of one for each landscape, with negative values indicating colder/ dryer than average years, and positive values indicating hotter/ wetter than average years. As previous studies suggest variable lag times between climate anomalies and disturbance signals determined from remote sensing[12,16], we tested variable lags ranging from 0 years (i.e., relating the climate anomaly of the current year to the disturbance in the current year) to 3 years (i.e., relating the climate anomaly 3 years prior to the disturbance in the current year). The lag best supported by the data within each cluster was identified using Akaike's Information Criterion (AIC)[16,55]. Furthermore, we allowed for an interaction term between precipitation and temperature to account for potential modulating effects between these two variables[56]. Using GLMMs enabled us to incorporate factors on different hierarchical levels into a combined modeling framework. In particular, TRI values were available at the pixel level, but remain constant across years. In contrast, the temperature and precipitation anomalies varied among years, but did not differ spatially in a study landscape. Hence, we used relative topographic complexity to explain the spatial variation in disturbance probability, while climate variability was related to temporal variation of disturbances in our model. In addition, the GLMM framework allowed us to account for random variation in the model intercept among study landscapes within a cluster. As sample sizes were very large (several millions of 30 m pixels), we randomly sampled 10% of the pixels per landscape. As disturbances were rare in many landscapes, we employed a case-control sampling design[57], that is we randomly down-sampled the absence class (no disturbance) to approximately the same size of the presence class (disturbance). This design has the advantage that model estimates are unbiased (with the exception of the intercept), and the intercept can easily be corrected using the known true proportion of disturbance presence/absence in the population. Finally, we compared three candidate models per cluster: a full model containing predictors of spatial (TRI) and temporal (precipitation and temperature anomalies as well as their interaction) variation, a model containing TRI only (spatial-only model), and a null model (containing only an intercept while maintaining the random effects structure of the GLMMs). Model comparison was done using the AIC and log- likelihood tests. For the model most strongly supported by the data we created response curves by drawing 9999 random simulation from the model (i.e., accounting for parameter and model uncertainty), as suggested by Gelman and Hill[58]. All models and calculations were implemented using the *lme4* package[59] (version 1.1-14) in R.

## Data availability

Data on forest disturbances were derived from the global forest change data set[24], available at https://earthenginepartners.appspot.com/science-2013-global-forest. Data on tree species traits were derived from the plant trait database TRY[49], available at https://www.try-db.org/TryWeb/Home.php (doi: 10.17871/TRY.3). Landscape-level data on ecological context variables was derived by means of a questionnaire (see Supplementary Table 5), and is published in full in Supplementary Table 1.

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

## Acknowledgements

A.S. and R.S. acknowledge support from the Austrian Science Fund (FWF) through START grant Y895-B25. C.S. acknowledges funding from the German Academic Exchange Service (DAAD) with funds from the German Federal Ministry of Education and Research (BMBF) and the People Programme (Marie Curie Actions) of the European Union's Seventh Framework Programme (FP7/2007–2013) under REA grant agreement Nr. 605728 (P.R.I.M.E.—Postdoctoral Researchers International Mobility Experience). T. D. acknowledges funding from the Fonds institutionnel de recherche de l'Université du Québec en Abitibi-Témiscamingue, the Natural Sciences and Engineering Research Council of Canada (NSERC), Tembec, and EACOM Timber Corporation. Á.G.G. was supported by FONDECYT 11150835. S.J.H. and T.T.V. acknowledge NSF Award 1262687. A.H. was partially supported by NSF (award #1738104). D.K. acknowledges support from the US NSF. D.L. was supported by an Australian Research Council Laureate Fellowship. A.S.M. was supported by the Environment Research and Technology Development Fund (S-14) of the Japanese Ministry of the Environment and by the Grants-in-Aid for Scientific Research of the Japan Society for the Promotion of Science (15KK0022). G.L.W.P. acknowledges support from a Royal Society of New Zealand Marsden Fund grant. S.L.S. acknowledges funds from the US Joint Fire Sciences Program (project number 14-1-06-22) and UC ANR competitive grants. M.S. and T.H. acknowledges support from the institutional project MSMT CZ.02.1.01/0.0/0.0/16_019/ 0000803. M.G.T. acknowledges funding from the University of Wisconsin-Madison Vilas Trust and the US Joint Fire Science Program (project numbers 09-1-06-3, 12-3-01-3, and 16-3-01-4). The study used data from the TRY initiative on plant traits (http://www.try-db.org). The TRY initiative and database is hosted, developed and maintained by J. Kattge and G. Boenisch (Max Planck Institute for Biogeochemistry, Jena, Germany). TRY is currently supported by Future Earth/bioDISCOVERY and the German Centre for Integrative Biodiversity Research (iDiv) Halle-Jena-Leipzig.

## Author contributions

A.S., C.S., and R.S. designed the study, analyzed the data and wrote the paper. B.B., A.W. D., T.D., I.D.-H., S.F., L.E.F., Á.G.G., S.J.H., B.J.H., H.S.H., T.H., A.H., T.K., D.K., D.L., A. S.M., J.M., J.P., G.L.W.P, S.L.S., M.S., M.G.T., and T.T.V. contributed data and commented on the manuscript.

## Additional information

**Competing interests:** The authors declare no competing interests.

