## [Peer Review File · Nature Communications]

Reviewers' comments:

Reviewer #1 (Remarks to the Author):

Summary

The authors present a quantitative biome-wide analysis of forest disturbance of the temperate forest biome for the years 2001-2014. For 50 sites with little human impact, disturbance was quantified with the annual forest disturbance maps of Hansen et al. (2013), and local expert knowledge of disturbance agents and forest type was acquired by questionnaire. Sites were classified by disturbance activity (low, moderate, or high) based on disturbance area proportion, patch size, and patch shape. The authors investigated how disturbance activity was related to a variety of factors: disturbance type, tree species, tree species composition, maximum tree height, wood density, and human influence on the landscape. Disturbance probability was predicted from topographic variability and previous-year climate anomalies for low, moderate, and high disturbance activity classes.

Significant differences existed between disturbance activity and many these factors. The relationships between disturbance activity and forest characteristics were mostly unsurprising, except for the relationship between forest species heterogeneity and disturbance activity, which was opposite of expected - more heterogeneous forests were associated with greater disturbance activity (Figure 2c). As expected, disturbance probability was generally positively associated with warmer and drier climate anomalies. Human influence affected disturbance patterns, except in highly disturbed areas.

Contribution

This paper should be of interest to readers interested in climate change and forest disturbance. Although the authors' findings are largely expected and similar to what has already been published for various disturbance types and locations, the scale of this biome-wide analysis makes this paper interesting and novel. It recalled to me the popular paper of Allen et al. (2010) who reviewed climate-induced tree mortality across the globe; this paper, however, gives a quantitative analysis rather than a review. The concept of classifying landscapes by disturbance activity is also novel, to my knowledge, and could be a useful concept that others could implement.

Comments

The paper is interesting, well written, concise, and convincing. Figures are simple and do a good job of summarizing and highlighting the authors' findings. The methods are well described, could be reproduced, and seem sound to me; I don't have experience with generalized linear mixed effects models and therefore cannot comment on those modeling specifics.

The meaning of the word 'patterns' in the title, and throughout the abstract and introduction was ambiguous to me upon first reading the manuscript. After studying the manuscript, I think "patterns" could be qualified by the word "spatial" or "landscape" in many instances, and will make the paper more clear to readers. In the cases where "pattern" refers to the climate analysis, maybe other wording could be used for clarity.

Line 203: Perhaps bark beetles were absent from the landscapes classified as low because only sites experiencing bark beetle outbreaks were included in the analysis? Maybe it is more appropriate to describe bark beetle activity as bark beetle outbreaks here and throughout the paper? This isn't necessarily a recommendation, just an observation.

Also, another similar distinction or limitation that might be worth noting, is that many disturbances, like low-level bark beetle activity or even low-severity fire, affect <50% of forest cover in 30 meter pixels. So the product of Hansen et al. (2013) is only detecting highly-disturbed pixels; low severity disturbance is ignored. If low severity disturbance covers a large area for a given site, estimates of percent forest area disturbed might be inaccurate. I'm not suggesting the

analysis needs to be altered, and I think the use of the product of Hansen et al. (2013) is appropriate for this analysis since it's standard across each site; I'm simply stating a limitation of 30 meter imagery that many readers will likely be familiar with, although perhaps it is worth noting briefly.

In summary, I think it is great paper that should be published.

Ben Bright

Reviewer #2 (Remarks to the Author):

Review for manuscript entitled: 'Patterns and drivers of recent disturbances across the temperate forest biome' submitted to the journal Nature Communications.

In the manuscript the authors report on forest disturbance within 50 landscapes across the globe and relate findings to temperature and precipitation anomalies. The paper is interesting and one of the first to synthesize findings across the global temperate biome. However, I think the paper is based upon a global (and blanket approach) classification system based upon Landsat data and it is hard to make sure that each disturbance is captured without any evaluation of the classification model. In addition, I wonder about the effectiveness of the separation of the 3 groups of landscapes using the Gaussian finite mixture models. In Figure S1 the disturbance density plots of the Low and Moderate group overlap for more than 50% making me wonder whether the classification of the 3 groups is not actually based upon forest patterns while the labeling of the 3 different groups by the authors suggests otherwise. I think the authors should address the following comments before the manuscript can be considered for publication in Nature Communications.

Major comments:

- Please provide an accuracy assessment of some of the sites, I am especially concerned that low-level dispersed tree mortality is not captured and is omitted in this analysis leading to underestimations of dispersed mortality.
- Re-think the labeling or methodology for separating the three groups (low, moderate, high) of disturbance activity, as the low and moderate group have significant disturbance severity overlap.
- Please indicate throughout the manuscript the you analyze protected as well as surrounding unprotected areas (not just protected areas).
- Whereas the relating the climate of the previous year to bark beetle-caused mortality makes sense, this does not hold true for fire disturbance (which are generally related to current year spring and summer precip and temp) and I am wondering what this effect is in the modeling of the climate variables. Would it be possible to run a lagged and a non-lagged model?
- I wonder how much of the landscape patch metrics differ excluding disturbance? I would expect some (most?) variability is just related to topography and landscape heterogeneity not related to stand-replacing disturbances. Is there a way to test for this?

Minor comments:

L55: "Increasing evidence suggests" —> awkward (evidence does not suggest, people do), please rephrase...

L56: "yet local variability in disturbance remains high" —> be more specific about what disturbance —> disturbance magnitude, disturbance severity?

L60: "Global" or "Local" —> you did not assess global disturbances.

L64: compared to their surroundings, but above you mention that you only investigated "50 protected landscapes".

L107: Add Hansen et al (2013) [25] to reference here.

L116: Remove "Here,".

L121-123: "Focusing our network of landscapes on protected areas" —> change to: "Focusing our network of landscapes on protected and surrounding areas"

Figure 1: Please make sure if the reference here [28], maybe this should be [43?] is correct.

L311: What does "tree species evenness" mean? Could you replace this with "tree species diversity" or something more intuitive?

L402: Replace "horizontal" with "spatial".

Reviewer #3 (Remarks to the Author):

The paper appropriately recognizes the value of global data in assessing environmental change as well as the appropriate use of local information in providing context. The results make sense and illustrate a global-scale application that provides statistical evidence of a climate signal affecting forest cover. I am not a statistician and the methods are dense and varied in terms of the statistical analyses. Unfortunately, I cannot provide meaningful feedback on the statistical methods, but assume they have been performed with rigor.

Specific comments:

Table S5 shows that assignment of disturbance factors resulted from an expert-generated list, meaning they are not assigned to pixels. I find this problematic for some of the statistical tests. For example, 19ha of forest disturbance occurred in Villarrica in Chile by 5 disturbance agents. This does not make sense. To what degree does the lack of specific assignation of disturbance agents proportional to actual disturbance likely change the narrative? Along these lines in Figure S3, prevalence of disturbance factor is compared. How is the ranking of generic disturbance factors incorporated into this and other analyses and results? I find this to be the weakest part of the analysis, the fact that you do not make a direct link to the mapped disturbances.

Upon first reading of the Abstract, the exact meaning of the following line was not clear to me:

64 "Yet, this signal of human land-use disappeared in areas with high recent natural disturbance activity, underlining the potential of climate-mediated disturbance change to transform forest landscapes."

It sounds like a significant result, but is not easily deciphered, as you have not mentioned land use

before. Maybe spell out that disturbance outside of PAs should be land use related, but is otherwise indicated for this category.

144-145 "We expected disturbances in protected areas to be generally smaller and more complex in shape (i.e. higher perimeter-area-ratio) compared to those outside of protected areas."

Why is this the case? Is this specific to temperate forests, as fire is so infrequent? Please add some explanatory text here, something like natural disturbances in temperate forests absent of fire consist of a, b, c, and are typically/normally smaller in extent than land use changes outside of protected areas. A reference would be good to have here as well.

257 "For landscapes with low disturbance activity we found a moderate negative effect of temperature anomaly, that is warmer than average temperatures decreased the probability of disturbance in the following year (Figure 4a)."

There is such huge variation here, I question the finding. I also find the narrative explaining why this is so to be lacking. Can you statistically and causally defend this?

290 "Our results thus highlight that while extensive disturbances such as massive beetle outbreaks or severe large-scale fires have garnered considerable attention from researchers and the public, many temperate forests have been dominated by small-scale disturbance events."

Depends on your definition of 'dominated.' The message made here implies researchers should be looking elsewhere, that their attention is misplaced, and I do not think that to be the case. Small-scale background disturbance could be taken as natural, whereas the regional scale disease and fire outbreaks are anomalous. That is why they garner so much attention. A good point to make is the importance of quantifying all dynamics, to create a consistent, comparable assessment of disturbance factors, something I take to be the primary purpose of this study. In the discussion, you appropriately mention the lack of a probability-based sample to do this, instead working where you are able and making the assumption that the chosen sites are representative of the biome as a whole.

303 "This pattern can partly be explained by the general life-history strategy of conifers and their extensive coevolution with disturbances³⁰, as many conifers are well adapted to either tolerate disturbances or swiftly recolonize disturbed areas."

What about alternative explanations related to climate-induced stress/mortality/fire risk? The intermountain west of the USA is mainly conifer and suffering high mortality, not simply experiencing a normal disturbance/recovery cycle.

305 "An unexpected finding was that the most abundant tree species was less dominant in landscapes with high disturbance activity compared to those with low and moderate disturbance activity, suggesting higher tree species evenness in high disturbance landscapes."

In general, when presenting a finding, could you include a statistic to bolster the claim? Maybe referencing the relevant table is the way to go, but I would prefer a parenthetical with the relevant statistic. Concerning this finding in particular, where do I find the particular statistical proof? Table S2 is not referred to anywhere in the main text.

Following the preceding text on conifer-related loss, I am not sure how to interpret this finding. Are high conifer-dominant loss sites not also most abundant tree species sites? Some bolstering and/or clarification of this conclusion is needed.

318 "In landscapes affected by large-scale disturbances, however, the patterns between protected areas versus their surroundings were similar. This suggests that large natural disturbances can override the effect of human land use and dominate landscape patterns in forest ecosystems. As these events have been found to be particularly climate-sensitive (Fig. 4c), future conditions could produce more coarse grained landscape patterns (i.e., landscapes characterized by larger patch sizes) in temperate forests³⁴."

and

358 "Our results thus suggest that a warming climate could facilitate large-scale disturbances in temperate forest ecosystems in the future^{2,42}."

I suppose this is the main finding – that climate-related large-scale disturbances are indiscriminate with regards to land use and protection status. That is ok by me.

354 Here we provide evidence that high recent disturbance activity in temperate forest ecosystems across the globe was strongly related to the joint occurrence of warmer and drier than average climate conditions (H3).

I like the H1/2/3 introduction and later recall in the discussion. However, I would also like to be presented with specific statistical evidence, as I have mentioned before. The introduction of H3 mentions regression results that would confirm the hypothesis. Please, spell out those results when talking of 'evidence' that confirms the hypothesis.

386 Selection criteria were that the landscapes are protected (i.e., IUCN Cat. I and Cat. II), and have a minimum of 2,000 ha of contiguous forest area. Considering only protected areas allowed us to control for anthropogenic disturbances and focus our main analyses on natural disturbances.

Even Cat. I IUCN protected areas allow for forest management within them. In other words, you can see land use within many PAs. Can you assure the absence of such dynamics within your selected PAs? This is important.

398 "an event that reduces forest cover below a threshold of 50% per pixel"

Probably should simply state "a stand-replacement disturbance."

Reviewer #4 (Remarks to the Author):

I have now read the manuscript "Patterns and drivers of recent disturbances across the temperate forest biome" by Sommerfeld et al. Due to my background, I paid special attention to the statistical analyses used in the article. It is my opinion that the statistical approaches chosen by the authors are appropriate. They are based on standard methods, implemented in reputable software and reported in sufficient detail.

Reviewers' comments:

Reviewer #1 (Remarks to the Author):

Summary

The authors present a quantitative biome-wide analysis of forest disturbance of the temperate forest biome for the years 2001-2014. For 50 sites with little human impact, disturbance was quantified with the annual forest disturbance maps of Hansen et al. (2013), and local expert knowledge of disturbance agents and forest type was acquired by questionnaire. Sites were classified by disturbance activity (low, moderate, or high) based on disturbance area proportion, patch size, and patch shape. The authors investigated how disturbance activity was related to a variety of factors: disturbance type, tree species, tree species composition, maximum tree height, wood density, and human influence on the landscape. Disturbance probability was predicted from topographic variability and previous-year climate anomalies for low, moderate, and high disturbance activity classes.

Significant differences existed between disturbance activity and many these factors. The relationships between disturbance activity and forest characteristics were mostly unsurprising, except for the relationship between forest species heterogeneity and disturbance activity, which was opposite of expected - more heterogeneous forests were associated with greater disturbance activity (Figure 2c). As expected, disturbance probability was generally positively associated with warmer and drier climate anomalies. Human influence affected disturbance patterns, except in highly disturbed areas.

Contribution

This paper should be of interest to readers interested in climate change and forest disturbance. Although the authors' findings are largely expected and similar to what has already been published for various disturbance types and locations, the scale of this biome-wide analysis makes this paper interesting and novel. It recalled to me the popular paper of Allen et al. (2010) who reviewed climate-induced tree mortality across the globe; this paper, however, gives a quantitative analysis rather than a review. The concept of classifying landscapes by disturbance activity is also novel, to my knowledge, and could be a useful concept that others could implement.

Comments

The paper is interesting, well written, concise, and convincing. Figures are simple and do a good job of summarizing and highlighting the authors' findings. The methods are well described, could be reproduced, and seem sound to me; I don't have experience with generalized linear mixed effects models and therefore cannot comment on those modeling specifics.

Response: We thank Reviewer #1 for her/ his positive assessment of our work and the constructive comments on how to further improve our manuscript!

The meaning of the word 'patterns' in the title, and throughout the abstract and introduction was ambiguous to me upon first reading the manuscript. After studying the manuscript, I think "patterns" could be qualified by the word "spatial" or "landscape" in many instances, and will make the paper more clear to readers. In the cases where "pattern" refers to the climate analysis, maybe other wording could be used for clarity.

Response: Thank you for this comment. We have now carefully reconsidered our use of the term pattern in the context of the current study, and have also pondered the possibility of using some of the suggested alternatives. However, given that the investigation of patterns (in space and time) and the respective use of the term has a long-standing tradition in (landscape) ecology we have refrained from using a different, less established term here. However, we have addressed the Reviewers concern by giving a clear definition of what we mean by the term "patterns" in the context of our contribution. This has been added in line 81, which is the first instance the term appears in the text.

Line 203: Perhaps bark beetles were absent from the landscapes classified as low because only sites experiencing bark beetle outbreaks were included in the analysis? Maybe it is more appropriate to describe bark beetle activity as bark beetle outbreaks here and throughout the paper? This isn't necessarily a recommendation, just an observation.

Response: We agree that speaking of bark beetle outbreaks is more appropriate, as minor bark beetle activity and low severity bark beetle mortality is below our detection threshold, but is likely a component in many of the investigated landscapes. We have revised our manuscript accordingly, rewording to bark beetle outbreaks in all relevant instances (lines 213, 215, 216, and 401).

Also, another similar distinction or limitation that might be worth noting, is that many disturbances, like low-level bark beetle activity or even low-severity fire, affect <50% of forest cover in 30 meter pixels. So the product of Hansen et al. (2013) is only detecting highly-disturbed pixels; low severity disturbance is ignored. If low severity disturbance covers a large area for a given site, estimates of percent forest area disturbed might be inaccurate. I'm not suggesting the analysis needs to be altered, and I think the use of the product of Hansen et al. (2013) is appropriate for this analysis since it's standard across each site; I'm simply stating a limitation of 30 meter imagery that many readers will likely be familiar with, although perhaps it is worth noting briefly.

Response: We agree with Reviewer #1 that low and medium severity disturbances are per definition not included in our analysis. We have now revised the Material and methods section once more to

make this aspect of our data more clear (lines 445-446) and have also added the issue to the discussion to highlight possible limitations that arise from our underlying data to the reader (lines 377-386).

In summary, I think it is great paper that should be published.

Response: Thank you for this overall positive assessment of our work!

Reviewer #2 (Remarks to the Author):

Review for manuscript entitled: 'Patterns and drivers of recent disturbances across the temperate forest biome' submitted to the journal Nature Communications.

In the manuscript the authors report on forest disturbance within 50 landscapes across the globe and relate findings to temperature and precipitation anomalies. The paper is interesting and one of the first to synthesize findings across the global temperate biome. However, I think the paper is based upon a global (and blanket approach) classification system based upon Landsat data and it is hard to make sure that each disturbance is captured without any evaluation of the classification model. In addition, I wonder about the effectiveness of the separation of the 3 groups of landscapes using the Gaussian finite mixture models. In Figure S1 the disturbance density plots of the Low and Moderate group overlap for more than 50% making me wonder whether the classification of the 3 groups is not actually based upon forest patterns while the labeling of the 3 different groups by the authors suggests otherwise. I think the authors should address the following comments before the manuscript can be considered for publication in Nature Communications.

Response: We thank reviewer #2 for her/his comments. We agree that both the issue of evaluation as well as the delineation of clusters is important, and we have revised the manuscript with a particular focus on these issues (see our detailed responses below). We feel that these revisions have helped to substantially further improve and refine our manuscript!

Major comments:

- Please provide an accuracy assessment of some of the sites, I am especially concerned that low-level dispersed tree mortality is not captured and is omitted in this analysis leading to underestimations of dispersed mortality.

Response: Thank you for this suggestion. We'd like to point out that a thorough accuracy assessment for the disturbance data used here was already performed previously by Hansen et al. (2013). Using ground-true data they determined an overall accuracy of 99.6 %, with both low omission and commission errors for the forest loss class (< 15 %). The Hansen et al. (2013) data were further validated by independent groups across temperate forests, and these additional analyses generally confirmed the high accuracy of the product. For example, Buma and Barrett (2015) found an overall agreement of 91% when comparing the Hansen data to high-resolution imagery in southern Alaska. Furthermore, Borelli et al. (2016) validated the Hansen map across Europe and found an overall accuracy of 81%. We have now extended the discussion section to inform the reader about these evaluation exercises on the dataset used here (lines 381-386). In the light of this previous body of work on evaluation we have refrained from conducting further evaluations of the dataset here.

Nonetheless, we agree with Reviewer #2 on the issue and importance of low severity tree mortality. This type of mortality is – per definition – not included in our analysis, as disturbances are here defined as a severe canopy disturbance, meaning the complete mortality of all trees taller 5 m within a pixel. We have now revised the manuscript to state more clearly that our analyses focus on high severity canopy disturbance in order to avoid confusion for the reader (lines 120-121 and 445-446). Also, we have amended the discussion section, explicitly highlighting that the disturbance estimates presented here do not include low severity disturbance and understorey mortality, and are thus likely a conservative estimate of overall tree mortality (lines 377-386).

Re-think the labeling or methodology for separating the three groups (low, moderate, high) of disturbance activity, as the low and moderate group have significant disturbance severity overlap.

Response: Thank you for this suggestion. We agree that our initial display item for the three groups suggested a low separation between our clusters. However, based on our numerical analysis (means and 95% confidence intervals in Table 1), the overlap between the clusters is not statistically significant. We have now revisited the issue, and identified the problem in misleading kernel density smoothing applied in our initial figure. We have thus revised Supplementary Figure 1 to now present the raw distributions in terms of boxplot. In line with the numerical analysis this new visualization shows the clear separation of the three groups for all four indicators. In addition, we have included additional information on our clustering approach in the revised version of the manuscript. Specifically, we now show the BIC for different clustering approaches and numbers of clusters, highlighting that the three cluster chosen here are most strongly supported by the data (see new Supplementary Figure 1).

- Please indicate throughout the manuscript the you analyze protected as well as surrounding unprotected areas (not just protected areas).

Response: Thank you for this comment. It is indeed an important aspect of our work that we not solely focus on protected areas but also compare them to their surroundings. We have now updated the text to make clear to the reader that we focused on protected areas as well as on their surroundings (for testing our second hypothesis, lines 60 and 122-123).

- Whereas the relating the climate of the previous year to bark beetle-caused mortality makes sense, this does not hold true for fire disturbance (which are generally related to current year spring and summer precip and temp) and I am wondering what this effect is in the modeling of the climate variables. Would it be possible to run a lagged and a non-lagged model?

Response: We agree with the Reviewer #2 that time lags can be expected to vary depending on the dominant disturbance agent. As we deemed this to be a very important issue we have considerably extended our analysis in this regard, for each cluster testing regression models with variable time lags (from 0 years to 3 years), using AIC values to determine which time lag is most strongly supported by the data (Supplementary Figure 5). This new analysis supports the notion of the Reviewer that time lags might differ between disturbance activity clusters, with the lowest lag (0 years) found for the fire-dominated high activity cluster. In addition to adding an interesting facet regarding lags to our analysis this revision also improved the statistical fit of our model substantially, resulting in smaller confidence intervals (Figures 2a, 2b, 2c) and clearer relationships to climatic covariates (Figure 2b). We thus are grateful to Reviewer #2 for suggesting to look into the issue of time lags, as it considerably strengthened our analysis further!

- I wonder how much of the landscape patch metrics differ excluding disturbance? I would expect some (most?) variability is just related to topography and landscape heterogeneity not related to stand-replacing disturbances. Is there a way to test for this?

Response: We agree with the Reviewer that landscapes will differ already in the absence of disturbance (e.g., with regard to their species composition, productivity, etc.). We here, however, focused on describing recent disturbance patterns, characterizing disturbed patches with regard to their size, prevalence shape, and complexity. Consequently, describing disturbance patterns in general, and analyzing the focal indicators of our study in particular, is not possible when disregarding disturbances. We nonetheless agree with Reviewer #2 on the importance of topography, and note that we have controlled for effects of variable topography in our regression analysis via GLMMs (see lines 542-549).

Minor comments:

L55: “Increasing evidence suggests” —> awkward (evidence does not suggest, people do), please rephrase...

Response: We agree with Reviewer #2 and have reworded the sentence (line 55).

L56: “yet local variability in disturbance remains high” —> be more specific about what disturbance —> disturbance magnitude, disturbance severity?

Response: We agree with the Reviewer that it would be desirable to be more specific here. However, as the abstract is restricted to 150 words and the context of this particular sentence becomes more clear with the following sentences, we have refrained from adding a detailed elaboration on our indicators of variability at this point of the text.

L60: “Global” or “Local” —> you did not assess global disturbances.

Response: Correct! We have omitted the term in the revised version of the text (line 61).

L64: compared to their surroundings, but above you mention that you only investigated “50 protected landscapes”.

Response: We revised the text to make clear that the focus of our analyses is on protected areas *and* their surroundings (line 60)

L107: Add Hansen et al (2013) [25] to reference here.

Response: Revised as suggested (line 107 revised manuscript)

L116: Remove “Here,”.

Response: Revised as suggested (line 117 revised manuscript)

L121-123: “Focusing our network of landscapes on protected areas“ —> change to: “Focusing our network of landscapes on protected and surrounding areas”

Response: Revised as suggested (lines 122-123 revised manuscript)

Figure 1: Please make sure if the reference here [28], maybe this should be [43?] is correct.

Response: Thank you for this comment. Indeed, there was a problem with our citation management software here. This is now fixed and the correct reference given – thanks for spotting this!

L311: What does “tree species evenness” mean? Could you replace this with “tree species diversity” or something more intuitive?

Response: Diversity in ecosystems has many dimensions, with the evenness of the species distribution being one important indicator. Evenness indicates the inequality in abundance of the occurring species, and thus is an important complementary indicator to species richness. As we here in fact use an indicator of evenness in our analysis we have retained the term also in the revised version of the analysis. We agree that substituting with diversity would improve the flow of the text, but it would also be somewhat less precise, hence our decision to retain the term evenness. In order to make more clear for the reader what we mean by evenness in this context we have revised the previous sentences to make more clear that we here focus on the relative dominance of the single most prevalent tree species.

L402: Replace “horizontal” with “spatial”.

Response: Revised as suggested (line 445 revised manuscript)

Reviewer #3 (Remarks to the Author):

The paper appropriately recognizes the value of global data in assessing environmental change as well as the appropriate use of local information in providing context. The results make sense and illustrate a global-scale application that provides statistical evidence of a climate signal affecting forest cover. I am not a statistician and the methods are dense and varied in terms of the statistical analyses. Unfortunately, I cannot provide meaningful feedback on the statistical methods, but assume they have been performed with rigor.

Response: We thank Reviewer #3 for her/ his overall positive assessment of our manuscript!

Specific comments:

Table S5 shows that assignment of disturbance factors resulted from an expert-generated list, meaning they are not assigned to pixels. I find this problematic for some of the statistical tests. For example, 19ha of forest disturbance occurred in Villarrica in Chile by 5 disturbance agents. This does not make sense. To what degree does the lack of specific assignment of disturbance agents proportional to actual disturbance likely change the narrative? Along these lines in Figure S3, prevalence of disturbance factor is compared. How is the ranking of generic disturbance factors incorporated into this and other analyses and results? I find this to be the weakest part of the analysis, the fact that you do not make a direct link to the mapped disturbances.

Response: We thank Reviewer #3 for this comment. We agree that the large number of disturbance agents combined with the lack of pixel-scale data could result in misleading results. We have now revised our analysis to only include the two most important disturbance agents per landscape as per the reporting of our local experts. This revised analysis has improved the signal-to-noise ratio of our analysis, but has not changed the overall picture regarding the relative importance of disturbance agents in the three disturbance activity clusters (see the revised Fig. 2a).

Upon first reading of the Abstract, the exact meaning of the following line was not clear to me:

64 “Yet, this signal of human land-use disappeared in areas with high recent natural disturbance activity, underlining the potential of climate-mediated disturbance change to transform forest landscapes.”

It sounds like a significant result, but is not easily deciphered, as you have not mentioned land use before. Maybe spell out that disturbance outside of PAs should be land use related, but is otherwise indicated for this category.

Response: We have revised the abstract as suggested, now making clear that the surrounding areas are indeed affected by human land use (line 65).

144-145 “We expected disturbances in protected areas to be generally smaller and more complex in shape (i.e. higher perimeter-area-ratio) compared to those outside of protected areas.”

Why is this the case? Is this specific to temperate forests, as fire is so infrequent? Please add some explanatory text here, something like natural disturbances in temperate forests absent of fire consist of a, b, c, and are typically/normally smaller in extent than land use changes outside of protected areas. A reference would be good to have here as well.

Response: We agree with the Reviewer that our hypothesis here needed further elaboration. We hypothesized smaller disturbances in protected areas for two main reasons: (1) Outside protected areas disturbances observed via satellite imagery are the result of both natural and human disturbances (which can have amplifying interactions, e.g. when strong winds affect edge trees of freshly created clearcuts), and (2) management-related biotic homogenization (e.g. with regard to tree species, forest structures) has the potential to increase the forest susceptibility to disturbances relative to natural ecosystem development. We have now added this information to the text to provide more context for the reader (lines 149-154).

257 “For landscapes with low disturbance activity we found a moderate negative effect of temperature anomaly, that is warmer than average temperatures decreased the probability of disturbance in the following year (Figure 4a).”

There is such huge variation here, I question the finding. I also find the narrative explaining why this is so to be lacking. Can you statistically and causally defend this?

Response: Our modeling here has been substantially revised based on a comment of Reviewer #2, now using variable time lags for each disturbance activity cluster. This has substantially improved the determination of our statistical models, narrowing the confidence bands and increasing the signal-to-noise ratio. In addition we have now added more statistical results (including p-values) to the results section describing our GLMM results, thus providing improved information regarding their statistical underpinning to the reader.

290 “Our results thus highlight that while extensive disturbances such as massive beetle outbreaks or

severe large-scale fires have garnered considerable attention from researchers and the public, many temperate forests have been dominated by small-scale disturbance events.”

Depends on your definition of ‘dominated.’ The message made here implies researchers should be looking elsewhere, that their attention is misplaced, and I do not think that to be the case. Small-scale background disturbance could be taken as natural, whereas the regional scale disease and fire outbreaks are anomalous. That is why they garner so much attention. A good point to make is the importance of quantifying all dynamics, to create a consistent, comparable assessment of disturbance factors, something I take to be the primary purpose of this study. In the discussion, you appropriately mention the lack of a probability-based sample to do this, instead working where you are able and making the assumption that the chosen sites are representative of the biome as a whole.

Response: We agree with the Reviewer that our initial sentence was potentially misleading. We have now revised the text to add the suggested statement on the importance of considering the full range of disturbance dynamics (lines 326-328).

303 “This pattern can partly be explained by the general life-history strategy of conifers and their extensive coevolution with disturbances³⁰, as many conifers are well adapted to either tolerate disturbances or swiftly recolonize disturbed areas.”

What about alternative explanations related to climate-induced stress/mortality/fire risk? The intermountain west of the USA is mainly conifer and suffering high mortality, not simply experiencing a normal disturbance/recovery cycle.

Response: We thank the Reviewer for this suggestion, and agree that the question of whether some of the disturbances here exceed a “normal” disturbance/ recovery cycle is a highly relevant question. Due to the short temporal coverage of our disturbance data (2001-2014), we are, however, not able to address this issue here. This limitation is explicitly mentioned for the reader in lines 387-390 of the revised version of the manuscript.

305 “An unexpected finding was that the most abundant tree species was less dominant in landscapes with high disturbance activity compared to those with low and moderate disturbance activity, suggesting higher tree species evenness in high disturbance landscapes.”

In general, when presenting a finding, could you include a statistic to bolster the claim? Maybe referencing the relevant table is the way to go, but I would prefer a parenthetical with the relevant

statistic. Concerning this finding in particular, where do I find the particular statistical proof? Table S2 is not referred to anywhere in the main text.

Response: Thank you for bringing this issue to our attention. We agree that the main findings of our analyses should be underpinned by the respective statistical analyses throughout the manuscript. We have thus added test statistics (coefficients, measures of uncertainty and corresponding p-values) to the text throughout the results section, in particular revising the section reporting on our GLMM results (see, for example, line 279 and 302). With regard to the particular result highlighted by the Reviewer we'd like to point out that the respective p-values are indicated in Fig. 2. Also, Supplementary Table 2, reporting on the detailed results of our tests, is now referred to in the main text in line 248.

Following the preceding text on conifer-related loss, I am not sure how to interpret this finding. Are high conifer-dominant loss sites not also most abundant tree species sites? Some bolstering and/or clarification of this conclusion is needed.

Response: It is in fact correct that the high disturbance activity cluster holds a higher share of coniferous species compared to other landscapes. The single most prevalent species in these high disturbance landscapes is, however, less dominant compared to other clusters (see Fig. 2c, and lines 236-237). We have now revised our discussion in order to make these findings more clear.

318 “In landscapes affected by large-scale disturbances, however, the patterns between protected areas versus their surroundings were similar. This suggests that large natural disturbances can override the effect of human land use and dominate landscape patterns in forest ecosystems. As these events have been found to be particularly climate-sensitive (Fig. 4c), future conditions could produce more coarse grained landscape patterns (i.e., landscapes characterized by larger patch sizes) in temperate forests³⁴.”

and

358 “Our results thus suggest that a warming climate could facilitate large-scale disturbances in temperate forest ecosystems in the future^{2,42}.”

I suppose this is the main finding – that climate-related large-scale disturbances are indiscriminate with regards to land use and protection status. That is ok by me.

Response: Yes, this is indeed one of our main findings! Thank you for this positive assessment of our work!

354 Here we provide evidence that high recent disturbance activity in temperate forest ecosystems across the globe was strongly related to the joint occurrence of warmer and drier than average climate conditions (H3).

I like the H1/2/3 introduction and later recall in the discussion. However, I would also like to be presented with specific statistical evidence, as I have mentioned before. The introduction of H3 mentions regression results that would confirm the hypothesis. Please, spell out those results when talking of 'evidence' that confirms the hypothesis.

Response: As mentioned already above we agree with the Reviewer #3 and have now included the statistical support for our results in the text throughout the results section. Specifically for H3, the test statistics can be found in lines 283 and 286).

386 Selection criteria were that the landscapes are protected (i.e., IUCN Cat. I and Cat. II), and have a minimum of 2,000 ha of contiguous forest area. Considering only protected areas allowed us to control for anthropogenic disturbances and focus our main analyses on natural disturbances.

Even Cat. I IUCN protected areas allow for forest management within them. In other words, you can see land use within many PAs. Can you assure the absence of such dynamics within your selected PAs? This is important.

Response: We agree with the Reviewer that even in strictly protected areas a minimum amount of direct human influence might be present. Therefore, we revised our wording to better reflect that human influences can't be completely ruled out even in protected areas (lines 146, 251, and 435). In addition, agreeing on the importance of this issue for the interpretation of our results, we have inquired with the local experts for our study landscapes about known human interventions in the context of forest disturbances. This new analysis showed that for 17 out of 50 study landscapes some human influence is still evident, mostly in the form of fire suppression around human infrastructures or salvage logging of disturbed sites. This issue is now clearly communicated in Supplementary Table 6.

398 "an event that reduces forest cover below a threshold of 50% per pixel"

Probably should simply state "a stand-replacement disturbance."

Response: We have revised the mortality definition in the text for clarity. As our analyses are strongly based on the data presented by Hansen et al. (2013), the revised text very closely follows their definition.

Reviewer #4 (Remarks to the Author):

I have now read the manuscript "Patterns and drivers of recent disturbances across the temperate forest biome" by Sommerfeld et al. Due to my background, I paid special attention to the statistical analyses used in the article. It is my opinion that the statistical approaches chosen by the authors are appropriate. They are based on standard methods, implemented in reputable software and reported in sufficient detail.

Response: Thank you for this positive assessment of our methodological approach.

REVIEWERS' COMMENTS:

Reviewer #1 (Remarks to the Author):

I have no further recommendations for improvement. I think the authors have adequately addressed my and other reviewers' concerns. I recommend that their manuscript be published.

Reviewer #2 (Remarks to the Author):

I thank the authors for improving the manuscript following the reviewers comments. I think the manuscript can be accepted to Nature Communication if the following minor edits are addressed.

L66: Remove "change"

Figure 4(a-c): The gray line and bars are really hard to distinguish.

L322: Change to "Variation in recent forest disturbance activity across the globe..."

L385: Remove "well".

Reviewer #3 (Remarks to the Author):

The comments have been sufficiently addressed.

REVIEWERS' COMMENTS:

Reviewer #1 (Remarks to the Author):

I have no further recommendations for improvement. I think the authors have adequately addressed my and other reviewers' concerns. I recommend that their manuscript be published.

Reviewer #2 (Remarks to the Author):

I thank the authors for improving the manuscript following the reviewers comments. I think the manuscript can be accepted to Nature Communication if the following minor edits are addressed.

L66: Remove “change”

Response: Revised as requested.

Figure 4(a-c): The gray line and bars are really hard to distinguish.

Response: We have now improved the contrast between the three lines in Fig. 4

L322: Change to “Variation in recent forest disturbance activity across the globe...”

Response: Revised as requested.

L385: Remove “well”.

Response: Revised as requested.

Reviewer #3 (Remarks to the Author):

The comments have been sufficiently addressed.